# COVID-19 Vaccination and Neurological Manifestations: A Review of Case Reports and Case Series

**DOI:** 10.3390/brainsci12030407

**Published:** 2022-03-18

**Authors:** Shitiz Sriwastava, Kanika Sharma, Syed Hassan Khalid, Sakhi Bhansali, Ashish K. Shrestha, Mahmoud Elkhooly, Samiksha Srivastava, Erum Khan, Shruti Jaiswal, Sijin Wen

**Affiliations:** 1Department of Neurology, Rockefeller Neuroscience Institute, West Virginia University, Morgantown, WV 26506, USA; syedhassankhalid@gmail.com; 2Department of Neurology, Wayne State University, Detroit, MI 48201, USA; samiksha_sami@hotmail.com; 3West Virginia Clinical and Translational Science Institute, Morgantown, WV 26506, USA; nickyjaiswal3@hotmail.com; 4Department of Biostatistics, West Virginia University, Morgantown, WV 26506, USA; 5Smt. N.H.L. Municipal Medical College, Ahmedabad 380006, India; drkanikasharma95@gmail.com; 6Calcutta National Medical College, Kolkata 700014, India; sakhiabhansali@gmail.com; 7Kathmandu Medical College Teaching Hospital, Kathmandu 44600, Nepal; sh_ashish@hotmail.com; 8Department of Neuropsychiatry, Minia University, Minia 61519, Egypt; elkhoolymahmoud27@outlook.com; 9B.J. Medical College and Civil Hospital, Ahmedabad 380016, India; erum2006@gmail.com

**Keywords:** COVID-19 vaccination, GBS, transverse myelitis, Pfizer BioNTech, Moderna, Janssen/Johnson and Johnson and AstraZeneca vaccine

## Abstract

Background: With 10 vaccines approved by the WHO and nearly 48% of people fully vaccinated worldwide, we have observed several individual case studies of neurological manifestations post-COVID-19 vaccination. Through this systematic review, we aim to discern these CNS and PNS manifestations following the COVID-19 vaccine to help produce methods to mitigate them. Methods: We conducted a thorough literature search of Google Scholar and PubMed from 1 December 2020 until 10 October 2021 and included all the case studies of COVID-19 vaccine-associated neurological side effects. The literature search and data analysis were performed by two independent reviewers according to prespecified inclusion and exclusion criteria using PRISMA. Results: The most common CNS manifestation was CVST (14.47%), found in females (64%) younger than 50 years (71%) after the first AstraZeneca dose (93%). Others included CNS demyelinating disorders (TM, ADEM, MS, NMOSD) (9.30%), encephalopathy/encephalitis (3.10%), and others (4.13%). The most common PNS manifestation was GBS (14.67%) found in males (71%) older than 50 years (79%), followed by Bell’s palsy (5.24%) and others (2.10%). Most occurred with the AstraZeneca (28.55%), Pfizer-BioNTech (9.18%), and Moderna (8.16%) vaccines. Nine (64%) out of the 14 patients with CVST died. However, most cases overall (42 out of 51) were non-fatal (82%). Conclusion: Several CNS and PNS adverse events have occurred post-COVID-19 vaccination, including CVST, GBS, and TM. High vigilance with early identification and treatment leads to better outcomes. Further studies with non-vaccinated controls might help in understanding the pathophysiologic mechanisms of these neurological manifestations following COVID-19 vaccination.

## 1. Introduction

The novel severe acute respiratory syndrome-coronavirus 2 (SARS-CoV-2) virus causing the COVID-19 pandemic has affected 448 million individuals worldwide and has cost over six million lives so far. Vaccine production has tremendously accelerated, leading to the development of multiple new and effective vaccines against COVID-19 in a relatively short period. As a result, the safety profile of these vaccines requires continuous and extensive monitoring, and any potential adverse effect must be duly reported and investigated. Since December 2020, different countries have conducted mass vaccinations. The recent World Health Organization (WHO) data show that a total of 10.7 billion vaccine doses have been administered across the globe, such that 63.4% of the world population has received at least one dose of a COVID-19 vaccine, with 18.05 million being administered each day [1].

Globally available vaccines against SARS-CoV-2 act by four different mechanisms to elicit an immune reaction against COVID-19. The mRNA-based vaccines (Pfizer-BioNTech and Moderna) introduce genetically engineered RNA or DNA to generate a viral protein capable of inducing an immune response to itself. Second, are the vector-based vaccines (Janssen/Johnson and Johnson and AstraZeneca) that use a virus to deliver the SARS-CoV-2 genome inside cells. These infected cells can then synthesize such antigenic proteins, against which the body safely generates an immune response. The third kind is a protein-based vaccine (Sputnik V) that uses the spike protein or its fragments for inciting an immune response. Lastly, is the inactivated/attenuated viral vaccine (Sinopharm/Sinovac-CoronaVac) that triggers the immune system by presenting a killed or weakened COVID-19 virus [2,3,4].

Neurological complications are reported in several vaccines, with SARS-CoV-2 being no exception. Neurological complications have been reported in several vaccines. These includes seizures and hypotonic/hypo responsive episodes following pertussis vaccination, meningoencephalitis (Japanese encephalitis vaccine), and Guillain–Barré syndrome and giant cell arteritis (influenza vaccine). Vaccine hesitancy and misinformation have added to the struggles of this global pandemic. Reports have shown neurological complications of SARS-CoV-2 vaccines are rare [5,6]. For instance, the phase 3 clinical trials of AstraZeneca reported a few cases of transverse myelitis (TM). This review enumerates post-marketing neurological side effects with the seven most commonly available vaccines, namely, Pfizer-BioNTech, Moderna, COVAXIN, AstraZeneca, Sinovac-CoronaVac, Janssen, and Sputnik V [7,8,9]. 

Constituents of each vaccine include an antigen, a delivery system, and an adjuvant; post-vaccine adverse effects can be caused by any of these components. Several pathogenic mechanisms have been described to ascertain how vaccines are associated with neurological complications. Mechanisms such as molecular mimicry, aberrant immune reactions, and neurotoxicity have been attributed to these complications [2,3]. There have been reports of mild neurological symptoms that include dizziness, myalgia, muscle spasms, headache, and paresthesia. A small number of case reports have demonstrated neurological manifestations ranging from Guillain–Barré syndrome (GBS), transverse myelitis, facial nerve palsy, and cranial nerves neuropathies [10,11,12,13]. Severe neurological manifestations, such as acute disseminated encephalomyelitis (ADEM), cerebral venous sinus thrombosis (CVST), and stroke, have been reported in the Vaccine Adverse Event Reporting System (VAERS) related to the Pfizer-BioNTech, Moderna, and J&J/Janssen COVID-19 vaccines [14,15] (Table 1).

Our aim was to present a thorough literature review of post-COVID-19 vaccinations related to the central nervous system (CNS) and peripheral nervous system (PNS) manifestations. This would equip practitioners to make more informed decisions and decide which patients (if any) are at risk for these adverse events. 

## 2. Methods

### 2.1. Study Selection and Criteria 

Using the keywords “COVID-19 Vaccine and CNS” complication, “COVID-19 vaccine and PNS complication”, and “SARS-CoV-2 Vaccine and neurological manifestation”, we searched databases including PubMed and Google Scholar from 1 December 2020 to 30 October 2021. Two reviewers independently performed the literature search. Review articles and consensus statements were excluded from the analysis. We used the Preferred Reporting Items for Systematic Reviews and Meta-Analyses (PRISMA) for the display of inclusions and exclusions [16]. On the basis of our search criteria, we found the following numbers of articles from PubMed (*n* = 98) and Google Scholar (*n* = 153). Amongst all, 79 were identified as duplicates. Finally, we screened 172 articles for title and abstracts and reviewed full-text literature per our study objective after removing 65 articles that were either missing clinical information or did not meet our study objective. We included 39 articles for review for quantitative analysis (Figure 1).

### 2.2. Inclusion Criteria 

The inclusion criteria for the published studies were: (1) patient age ≥18 years; (2) history of neurological adverse event(s) post-COVID-19 vaccination; (3) established neurological diagnosis in the patients with COVID-19 vaccination; (4) neuroimaging findings of CNS and PNS complications not accounted for by another neurological process.

### 2.3. Exclusion Criteria

The exclusion criteria for the studies were: (1) duplicate studies that involved repetition of cases; (2) studies in languages other than English; (3) studies with no individual data on and/or fatality of COVID-19 vaccination, and (4) studies with missing clinical information. 

Cases were categorized into the following groups: (i) CNS inflammatory disorders including multiple sclerosis (MS)/MS exacerbation, acute disseminated encephalomyelitis (ADEM), transverse myelitis (TM), and neuromyelitis optica spectrum disorder (NMOSD); (ii) encephalitis and encephalopathy; (iii) GBS and its variants; (iv) Bell’s palsy; (v) ischemic disorders including ischemic stroke (both large and small vessel) associated with cerebral venous sinus thrombosis (CVST); and (vi) others, including small fiber neuropathy, phantosmia, Tolosa–Hunt syndrome, seizure disorders, and acute aseptic meningoencephalitis with Sweet syndrome. These cases were further classified on the basis of underlying pathophysiology into central nervous system (CNS) and peripheral nervous system (PNS) manifestations. Neurological adverse events that were either few in number or considered minor and, thus, excluded from this analysis were as follows: no specific neurologic diagnosis, neuromuscular disorders, hemorrhagic disorders (ICH), ischemic strokes of cardio-embolic or unknown origin, neuropathic pain, dizziness, and skeletal muscle injury. Moreover, we carefully checked the references of the included studies to cover all possible eligible studies. 

### 2.4. Quality Assessment 

The critical appraisal checklist for case reports provided by the Joanna Briggs Institute (JBI) was used to assess the overall quality of case series and case reports [17]. 

### 2.5. Data Acquisition 

From the selected studies, we extracted the following data for our analysis: study type, date of publication, age, gender, and clinical presentation following COVID-19 vaccination.

### 2.6. Data Analysis 

We conducted a statistical analysis to report demographic characteristics such as age, gender, severity, and outcomes of all COVID-19 post-vaccination cases presenting with neurological manifestations. In particular, we conducted the following analyses: (1) age; (2) gender; (3) outcome (including fatality); (4) type of COVD-19 vaccine; and (5) neurological manifestations. The chi-squared test and Wilcoxon rank-sum test were used in the data analysis for categorical and continuous variables, respectively. All statistical tests were two-sided, and a *p*-value < 0.05 implied the statistical significance in this study. Analysis was performed using SAS (version 9.2) and R software (version 3.6.3, R Foundation, Vienna, Austria). 

## 3. Results

In this review, we only included results describing neurological manifestations from reports where individual data were complete, leading us to a total of 39 studies and 51 patients. We summated our findings in Table 2 and Table 3. 

Out of the 51 individuals whose data was procured, 28 (55%) of them received the AstraZeneca vaccine, 9 (18%) received Pfizer-BioNTech, 8 (16%) received Moderna, 2 (4%) received Janssen, 2 (4%) received Sinovac-CoronaVac, and 1 person each received Sputnik and COVAXIN (2%) (Table 2). About half were male (51%) and were 50 years or older (55%) with no existing comorbid conditions (43%). Among those who did have comorbidities, hypertension (HTN) was the most common (22%), followed by diabetes mellitus (DM) type 2 (12%) and others (including CKD, CAD, hyperlipidemia, etc.) (combined 33%). 

### 3.1. The Associated Adverse Events

The associated adverse events were as follows: GBS (and its variants) (n = 14, 27%), Bell’s palsy (*n* = 5, 10%), ischemic disorders associated with CVST (*n* = 14, 27%), CNS demyelinating disorders (TM, ADEM, MS exacerbation, NMOSD) (*n* = 9, 18%), encephalitis/encephalopathy (*n* = 3, 6%), and others (*n* = 6, 12%), which were comprised of seizure disorders (*n* = 2), small fiber neuropathy (*n* = 1), phantosmia (*n* = 1), Tolosa–Hunt syndrome (*n* = 1), and acute aseptic meningoencephalitis with Sweet syndrome (*n* = 1) (Table 3). We further analyzed the data on the basis of outcome and vaccine type.

### 3.2. CNS Findings

The most common CNS manifestation was ischemic disorders related to cerebral venous sinus thrombosis (CVST) (Table 3). Out of 14 cases, 13 (93%) occurred after the first dose of the AstraZeneca vaccine, and 1 (7%) after the second dose of Moderna. The majority were younger than 50 years (71%) and female (69%), with nine (64%) fatalities (Table 3). HTN was present in two (14%) patients and DM type 2 in one (7%) patient; some had risk factors for CVST, such as a history of smoking (*n* = 2), and OCP (*n* = 1)/hormone replacement therapy (HRT) use (*n* = 1) (data unavailable for one case). The median duration of symptom onset from the time of vaccination was nine days, wherein headache was the most common presentation (80%), followed by hemiparesis (46%), visual symptoms (33%), and seizures (27%) (Appendix A). Most patients were managed medically using anticoagulants (*n* = 8), IV steroids (*n* = 4), IVIG (*n* = 4), and platelet transfusions (*n* = 4). Four (27%) patients required decompressive craniectomy, and one (7%) underwent an endovascular thrombectomy. Treatment was not performed for three cases due to poor prognosis and was not available for one case. 

CNS demyelinating disorders were the next most common manifestation, of which nine cases were found, including TM (*n* = 5), NMOSD (*n* = 2), ADEM (*n* = 1), and one case of MS relapse in a previously diagnosed patient with RRMS (Appendix A). Most of these were also females (67%) younger than 50 years (63%) (age not available for one case) who received the AstraZeneca vaccine (44%) (Table 3). All occurred after the first dose with a median time to symptom onset of 4 days post-vaccination. Additionally, all cases were non-fatal that responded to corticosteroid therapy; no cases were found with the COVAXIN and Janssen vaccines (Table 3 and Appendix A). 

Lastly, encephalopathy and encephalitis were reported in three patients, all of whom experienced symptoms within one week following the first dose of the Moderna vaccine (Table 3). Two patients experienced non-convulsive focal status epilepticus and were diagnosed with encephalopathy treated with lorazepam and levetiracetam. One patient was diagnosed with autoimmune encephalitis and treated with IV steroids; no deaths occurred (Appendix A). 

### 3.3. PNS Findings

GBS (and its variants) was the most common PNS manifestation (n = 14), found largely in males (71%) who were 50 years or older (79%) (Table 3). The incidence was highest after the first dose of the AstraZeneca vaccine (64%), followed by the second dose of the Pfizer-BioNTech (22%), and a single dose of the Janssen vaccine (14%). None were found linked to the Sinovac-CoronaVac, COVAXIN, and Sputnik vaccines. One patient previously diagnosed with CIDP had a relapse 17 days after the first dose of the AstraZeneca vaccine, which improved on treatment (Appendix A). Apart from the typical features present in most of these patients, we also found five cases of bifacial diplegia with paresthesias (BFP) variant of GBS and one with papilledema as an atypical onset (Appendix A). Outcomes were favorable, with no fatalities reported and IVIG being the treatment of choice (78%), followed by corticosteroids (36%) (Appendix A). 

Bell’s palsy was seen in five cases associated with the Pfizer-BioNTech (*n* = 3), COVAXIN (*n* = 1), and the Sinovac-CoronaVac vaccines (*n* = 1) (Table 3). One patient (20%) had sequential facial palsy following each dose of the Pfizer-BioNTech vaccine, and another (20%) had a history of Bell’s palsy before vaccination (Appendix A). All patients showed improvement after receiving oral steroids and topical therapy (Appendix A). 

A single case of small fiber neuropathy with subacute onset was reported one week after inoculation with the Pfizer-BioNTech vaccine, presenting as bilateral dysesthesia in the extremities, without other symptoms. Similarly, one radiologically confirmed diagnosis of neurologic phantosmia was also seen following the second dose of the Pfizer-BioNTech vaccine (Appendix A).

## 4. Discussion

This is the most recent record documenting various neurological adverse effects post-vaccination with the six approved and most commonly available COVID-19 vaccines. However, due to unequal distribution and administration, certain vaccines have been overrepresented in some countries compared to others. As of September 2021, Pfizer-BioNTech (BNT162b2) and AstraZeneca (ChAdOx1nCoV-19) were the most frequent vaccines administered (with 35.4 and 32.4 hundred million doses distributed, globally). The same source reports that the AstraZeneca vaccine is approved in 120 countries/regions, the highest of any vaccine. The Sinovac-CoronaVac vaccine is approved in 40 countries/regions (least of the six studied) [18]. Much like the parainfectious and post-infectious neurological effects of COVID-19, the neurological adverse effects of the vaccine are not unique to SARS-CoV-2. 

Research in the past has highlighted that adverse events following immunization (AEFI) are very rare, and as more people are vaccinated, more adverse events (AE) will be reported [19]. This could explain why 55% of the findings in this review were associated with the most commonly approved vaccine, AstraZeneca (Table 1). Being cognizant of this, the reported AE are discussed here after dividing them into CNS and PNS manifestations. The VAERS and MHRA keep a record of every such event following the COVID-19 vaccine. Both their reports have shown ischemic cerebrovascular accidents to be the most frequently seen AE [20,21,22]. Similarly, the most common AE in this review was ischemic events related to cerebral venous sinus thrombosis (CVST). 

### 4.1. CNS Complications

Of the 30 post-vaccination CNS manifestations we found, ischemic stroke due to CVST was the most common (14, 47%) (Table 3 and Appendix A). Among these, seven had at least one comorbid condition known to increase the risk factor of stroke, including HTN, type 2 DM, smoking, and OCP use. Immune-driven thrombosis causing CVST and thrombocytopenia are known adverse events following vaccination [23,24,25,26]. The first few cases of vaccine-induced thrombocytopenia and prothrombotic syndrome (VITT) were reported with the AstraZeneca vaccine, where pathological antibodies against platelet factor 4 (PF4) were found [27]. Likewise, 93% of cases in this review also received the AstraZeneca vaccine (64% were females, 71% were younger than 50 years). 

The VITT cases in this review mirrored recent trends where patients exposed to the J&J/Janssen (Ad26.COV2.S) vaccine had higher chances of developing thrombosis with thrombocytopenia syndrome (TTS) [28]. These findings explain the CDC’s recent (December 2021) decision of preferring Moderna and Pfizer vaccines above J&J in patients prone to hypercoagulable conditions [29]. Moreover, TTS is known to preferentially affect young females [30]. There were 12 US cases of CVST with thrombocytopenia following the Janssen vaccine were reported to the VAERS from 2 March to 21 April 2021 [30]. Both the AstraZeneca and Janssen vaccines are adenoviral vector-based vaccines; other vector-based vaccines, such as Sinovac and Sputnik, have not had TTS-like events. There are established guidelines to diagnose and treat patients with vaccine-induced thrombocytopenia and thrombosis (VITT) [31]. The latest recommendations by the American Society of Hematology suggest IVIG in such cases and advise against platelet transfusion. While the association between vaccines and prothrombotic states is being studied, it is important to remember that these side effects are rare and much less common than both CVST and ischemic stroke associated with the COVID-19 infection itself, as illustrated by a recent large epidemiological study [32]. To establish an increased incidence of stroke in the vaccinated population, one must compare it with that in the non-vaccinated population/in the pre-pandemic era. A nation-wide study conducted in France in April 2021 on people aged 75 and above reported no increase in the incidence of stroke 14 days following each Pfizer-BioNTech vaccine dose [33]. Israeli and U.S. studies reported that persons receiving the BNT162b2 vaccine were not at increased risk of cerebrovascular events in the 42 and 21 days following vaccination, respectively (Appendix A) [34,35]. Conversely, intracranial hemorrhage (ICH) with a normal cerebral CT angiography has also been found, presenting as aphasia one week after the second dose of an unspecified mRNA-based SARS-CoV-2 vaccine [36].

The review also reports nine cases of CNS demyelinating disorders (Appendix A). Previous accounts of vaccine-induced immune activation and inflammation exacerbating CNS demyelinating in individuals with prior history are present. The vaccine may also act as a trigger for demyelination in cases with subclinical disease or a genetic risk factor. However, no reports to date have established causality between vaccination and subsequent CNS demyelination syndromes [37]. Studies have shown that these events were more likely to be coincidental. A large study following 555 patients and their status post-vaccination revealed the rates of patients with acute relapse were 2.1% and 1.6% following the first and second doses, respectively. This was similar to the rate in non-vaccinating patients during the corresponding period. Younger patients were more likely to have a relapse post-vaccination, as seen in our review as well (63% vs. 37%) [38]. Three cases (10%) of encephalopathy/encephalitis associated with the Moderna vaccine were also found in our report, the details of which are described (Appendix A).

### 4.2. PNS Complications

Guillain–Barré syndrome (GBS) (and its variants) was the most common PNS manifestation among the vaccine-associated neurological adverse events (67%) (Table 3 and Appendix A). The first published report of GBS following COVID-19 vaccination was in an elderly female who presented two weeks after the initial dose of the Pfizer-BioNTech vaccine [39]. Since then, a growing number of such cases have led to the Therapeutic Goods Administration (TGA) of Australia declaring GBS as “an adverse event of special interest” associated with the Vaxzevria (AstraZeneca) vaccine [40,41]. As per the WHO, increased reports have only been observed following the adenoviral vector-based vaccines (Janssen and AstraZeneca), but not with the mRNA COVID-19 vaccines (Pfizer-BioNTech and Moderna) [42]. For instance, as of 13 July 2021, there were 100 preliminary reports of GBS in the USA after receiving the Janssen vaccine and one death with 12.5 million vaccine doses administered. This informed the FDA revision of the Janssen COVID-19 vaccine recipient and provider fact sheets to include an increased risk of GBS, especially in the 42 days following this single dose [43]. Similarly, the European Medicine Agency (EMA) formally listed GBS as “a very rare side effect” of the Janssen vaccine and later recommended adding a warning the same as with the AstraZeneca vaccine [44].

Just like VITT, GBS is also an immune-mediated process and includes a wide variety of demyelinating conditions. Patients in our study were mostly elderly males who were newly diagnosed with GBS post-vaccination (except for one case of relapse) [45], with five cases of bifacial diplegia with paresthesias (BFP) variant of GBS; however, atypical presentations were also seen [46]. Outcomes were favorable, with no fatalities reported and IVIG being the treatment of choice (78%). The incidence was highest among those receiving the first dose of their AstraZeneca vaccine (64%), with a median time to symptom onset of 14.5 days (Appendix A). This finding correlates with the statements above and with the most recent population-based study in England, where 38 excess cases of GBS per 10 million persons was calculated among those immunized with the ChAdOx1nCoV-19 vaccine in the 1–28 day risk period. However, the overall risk of GBS after this vaccination (IRR, 2.04) was still significantly lower than that after a positive SARS-CoV-2 test (IRR, 5.25) [47]. Additionally, no increased risk was associated with the Pfizer-BioNTech vaccine, with further data concluding the safety of this mRNA vaccine, even in previously diagnosed cases of GBS [47,48]. We also found no cases of GBS linked to the Sputnik, COVAXIN, and Sinovac-CoronaVac vaccines. Of note, one of the cases included in our study is from the Janssen COVID-19 Vaccine Clinical Trial (where the incidence of GBS was equal in both the placebo and active arms) [49].

GBS after vaccination is rare and has historically been reported with many vaccines. Prior research has estimated one to two cases per million immunized persons, and a multitude of epidemiological studies have proven no causal associations with most vaccinations [50]. One theoretical explanation regarding the COVID-19 vaccines may include a cross-reaction between antibodies produced to the SARS-CoV-2 spike protein and the sialic acid-containing glycoproteins, considering that the spike protein can bind to cell surface sialic acids (including those on the ACE2 receptor) [50]. However, until causality is proven, the potential benefits of these vaccines continue to outweigh any potential risk of GBS [51]. The remaining post-COVID-19 vaccination PNS disorders (Bell’s palsy, small fiber neuropathy, and phantosmia) [33%] are mentioned above (Section 3) and in Appendix A.

Additionally, eight retrospective large cohort observational studies were not included in statistical analysis but were reported separately, as the substantial data for individual neuroimaging, severity, and fatality of post-COVID-19 vaccination were not available from these studies (Appendix A).

We explored further reviewed literature to report grouped patients that were not included in our study analysis. Appendix A displays the vaccine type, MRI findings, and associated neurological manifestations along with outcomes for post-COVID-19 vaccination for all the observational studies included in our review. In the eight observational studies, 7077 patients demonstrated neurological manifestation following COVID-19 vaccination (Appendix A). Among them, CNS manifestations (5128, 72.4%) were more common than PNS manifestations (1949, 27.5%), mostly caused by ChAdOx1 nCoV-19 (5241, 74%) followed by Pfizer (1902, 26%). Of all the CNS cases, the common manifestations were as follows: headache (2829, 55.1%); hemorrhagic stroke (918, 17.9%); SAH (459, 8.9%); CVST (424, 8.2%); encephalitis, meningitis, and myelitis (285, 5.5%); and CNS demyelination (213, 4.1%). Similarly, among the PNS cases, the common manifestations were as follows: facial palsy (1529, 78.4%), myasthenic disorder (193, 9.9%), GBS (187, 9.5%), and sensorineural hearing loss (40, 2%). Most of the outcome of the patients were favorable, with 5843 cases (82.5%) being non-fatal and only 1234 cases (17.4%) being fatal (Appendix A).

### 4.3. Neuroimaging Findings

A sub-analysis of the reported imaging modalities was performed, consisting of various MRI/CT results (Appendix A). Data were obtained from 39 out of the 51 cases. Of these, the most common abnormal findings were ischemic stroke/infarction including lacunar infarct, CVST (14, 35.8%) associated with hemorrhagic stroke/ICH, SAH (13, 33.3%), encephalitis/encephalopathy (4, 10.2%), and microhemorrhages (2, 5.1%). In transverse myelitis, T2—hyperintense lesions and T1—gadolinium contrast enhancement were seen at the cervico-thoracic cord level (Appendix A).

Among the PNS manifestations, the most common neuroimaging finding was enhancement of the cauda equina nerve roots (*n* = 4, 10.2 %) seen with GBS. Bell’s palsy (another frequent presentation) had normal imaging results (Appendix A).

This is one of the first comprehensive reviews detailing the clinical presentation, management, and outcomes of various neurological adverse events (CNS and PNS) following the six most commonly available COVID-19 vaccines. Our study should be considered in light of several limitations. Cases included in this review were identified through an extensive search of databases using a systematic search strategy. However, we acknowledge the possibility of missing out on novel studies, owing to the highly active research within this field. Compelling evidence on such neurological manifestations associated with the COVID-19 vaccination continues to emerge and serves as a strong foundation for conducting this review. Lastly, we wish to highlight the lack of analogous case studies (not associated with the COVID-19 vaccination) from respective institutions since that would provide us an appropriate comparison between the clinical nature of these conditions and enhance our understanding of their underlying pathophysiology. We believe this may be a future indication from our study warranting further investigations.

## 5. Conclusions

The COVID-19 pandemic has tested the grit of medical sciences. Our understanding of this disease and its pathophysiology has gone hand in hand with our efforts to prevent/treat it. The development and administration of these vaccines in a span of two years is proof of rapidly evolving evidence-based medicine working in synch with the healthcare system. Neurological manifestations following the SARS-CoV-2 vaccination have been reported, and although they are few in number, healthcare professionals should be alert to their presentation as a high vigilance and rapid response to these events are the need of the hour. Further investigations are required to establish a definitive causal association with the currently recommended vaccines. Until then, the benefit of protection against COVID-19 for both individuals and society is far greater than the hypothesized risk of these adverse events. 

## Figures and Tables

**Figure 1 brainsci-12-00407-f001:**
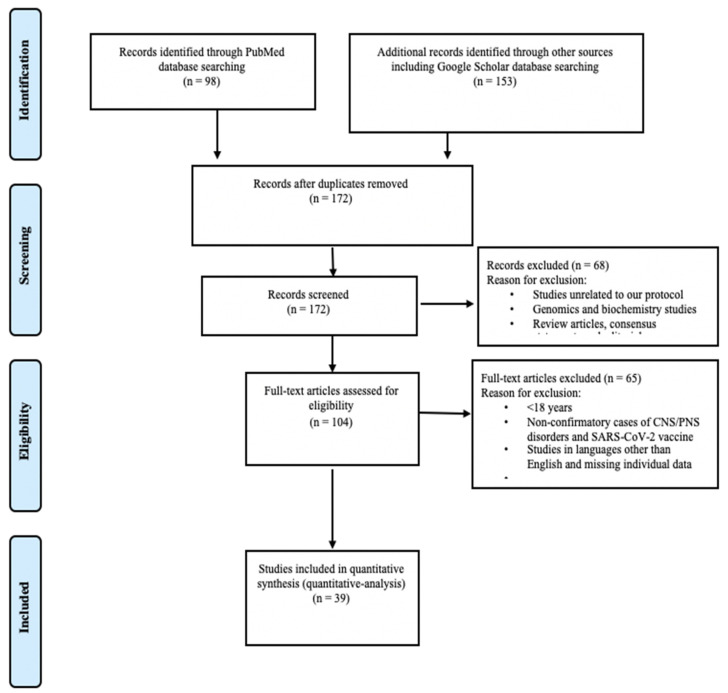
Preferred Reporting Items for Systematic Reviews and Meta-Analyses (PRISMA) Flow Diagram.

**Table 1 brainsci-12-00407-t001:** Medicines and Healthcare products Regulatory Agency (MHRA) data as of 12/15/21. Vaccine adverse events reporting system (VAERS) data as of 12/15/21.

(**a**)			
**Neurological Complications**	**Pfizer-BioNTech** **BNT162b**	**Moderna** **mRNA-1273**	**AstraZeneca Vaccine**
Cerebrovascular accident	423	18	1285
Ischemic stroke	56	1	157
Hemorrhagic stroke	11	0	46
Guillain–Barré syndrome	72	7	473
Transverse myelitis	28	2	105
Bell’s palsy	527	54	608
Cerebral venous sinus thrombosis	52	3	211
Optic neuritis	30	3	60
(**b**)			
**Neurological Complications**	**Pfizer-BioNTech** **BNT162b**	**Moderna** **mRNA-1273**	**Janssen**
Cerebrovascular accident	1425	1394	406
Ischemic stroke	196	135	56
Hemorrhagic stroke	59	45	15
Guillain–Barré syndrome	368	266	270
Transverse myelitis	105	84	28
Bell’s palsy	1813	1437	194
Cerebral venous sinus thrombosis	74	72	60
Optic neuritis	60	46	9

**Table 2 brainsci-12-00407-t002:** General characteristics of patients (*n* = 51) with neurological manifestations following different COVID-19 vaccines.

DEMOGRAPHIC DATA	VACCINE TYPES, n (%)							
	Pfizer-BioNTech	Moderna	Janssen	AstraZeneca	Sinovac-CoronaVac	Sputnik V	COVAXIN	All
	*n* = 9	*n* = 8	*n* = 2	*n* = 28	*n* = 2	*n* = 1	*n* = 1	*n* = 51
**AGE**								
<50	1(11)	4(50)	1(50)	15(54)	1(50)	1(100)	0(0)	23(45)
≥50	8(89)	4(50)	1(50)	13(46)	0(0)	0(0)	1(100)	27(55)
**GENDER**								
M	3(33)	5(62)	1(50)	16(57)	0(0)	0(0)	1(100)	26(51)
F	6(67)	3(38)	1(50)	12(43)	2(100)	1(100)	0(0)	25(49)
**COMORBIDITIES ***								
None	4(44)	2(25)	1(50)	13(46)	1(50)	0(0)	1(100)	22(43)
DM	1(11)	2(25)	0(0)	3(11)	0(0)	0(0)	0(0)	6(12)
HTN	4(44)	1(13)	0(0)	6(21)	0(0)	0(0)	0(0)	11(22)
Others	4(44)	3(37)	1(50)	7(25)	1(50)	1(100)	0(0)	17(33)
**TREATMENTS ****								
Steroids (Oral)								
Yes	4(44)	3(38)	0(0)	11(39)	2(100)	0(0)	1(100)	21(41)
No	5(56)	5(62)	2(100)	17(45)	0(0)	1(100)	0(0)	30(59)
Steroids (IV)								
Yes	0(0)	2(25)	0(0)	3(11)	0(0)	1(100)	0(0)	6(12)
No	9(100)	6(75)	2(100)	25(89)	2(100)	0(0)	1(100)	45(88)
Aspirin								
Yes	0(0)	0(0)	0(0)	0(0)	0(0)	0(0)	0(0)	0(0)
No	9(100)	8(100)	2(100)	28(100)	2(100)	1(100)	1(100)	51(100)
Clopidogrel								
Yes	0(0)	0(0)	0(0)	0(0)	0(0)	0(0)	0(0)	0(0)
No	9(100)	8(100)	2(100)	28(100)	2(100)	1(100)	1(100)	51(100)
Anticoagulants								
Yes	0(0)	1(13)	0(0)	6(21)	0(0)	0(0)	0(0)	7(14)
No	9(100)	7(87)	2(100)	22(79)	2(100)	1(100)	1(100)	44(86)
Plasmapheresis								
Yes	0(0)	2(25)	0(0)	1(4)	0(0)	0(0)	0(0)	3(6)
No	9(100)	6(75)	2(100)	27(96)	2(100)	1(100)	1(100)	48(94)
IVIG								
Yes	3(33)	0(0)	2(100)	11(39)	0(0)	0(0)	0(0)	16(31)
No	6(67)	8(100)	0(0)	17(61)	2(100)	1(100)	1(100)	35(69)
Antiepileptics								
Yes	1(11)	3(37)	0(0)	4(14)	0(0)	0(0)	0(0)	8(16)
No	8(89)	5(63)	2(100)	24(86)	2(100)	1(100)	1(100)	43(84)
Antivirals								
Yes	1(11)	1(12)	0(0)	0(0)	0(0)	0(0)	0(0)	2(4)
No	8(89)	7(88)	2(100)	28(84)	2(100)	1(100)	1(100)	49(96)
**FATALITIES**								
Yes	0(0)	0(0)	0(0)	9(32)	0(0)	0(0)	0(0)	9(18)
No	9(100)	8(100)	2(100)	19(68)	2(100)	1(100)	1(100)	42(82)

* associated with DM, HTN; ** treatment includes oral steroid, antiplatelet, PLEX, IVIG.

**Table 3 brainsci-12-00407-t003:** Comparison of neurological manifestations (*n* = 51) against age, outcomes, and types of COVID-19 vaccines.

COMPLICATIONS		VACCINE TYPES, *n* (%)							
**Ischemic Disorders ***		**Pfizer-BioNTech**	**Moderna**	**Janssen**	**AstraZeneca**	**Sinovac-CoronaVac**	**Sputnik V**	**COVAXIN**	**Total**
		***n* = 0**	***n* = 1**	***n* = 0**	**n = 13**	***n* = 0**	***n* = 0**	***n* = 0**	***n* = 14**
AGE	<50	0	1(100)	0	9(69)	0	0	0	10(71)
	≥50	0	0(0)	0	4(31)	0	0	0	4(29)
GENDER	M	0	1(100)	0	4(31)	0	0	0	5(36)
	F	0	0(0)	0	9(69)	0	0	0	9(64)
FATAL	Yes	0	0(0)	0	9(69)	0	0	0	9(64)
	No	0	1(100)	0	4(31)	0	0	0	5(36)
**CNS Demyelinating disorders ****		**Pfizer-BioNTech**	**Moderna**	**Janssen**	**AstraZeneca**	**Sinovac-CoronaVac**	**Sputnik V**	**COVAXIN**	**Total**
		***n* = 1**	***n* = 2**	***n* = 0**	***n* = 4**	***n* = 1**	***n* = 1**	***n* = 0**	***n* = 9**
AGE	<50	0(0)	1(50)	0	3(75)	NA	1(100)	0	5(63)
	≥50	1(100)	1(50)	0	1(25)	NA	0(0)	0	3(37)
GENDER	M	0(0)	0(0)	0	3(75)	0(0)	0(0)	0	3(33)
	F	1(100)	2(100)	0	1(25)	1(100)	1(100)	0	6(67)
FATAL	Yes	0(0)	0(0)	0	0(0)	0(0)	0(0)	0	0(0)
	No	1(100)	2(100)	0	4(100)	1(100)	1(100)	0	9(100)
**Encephalopathy and Encephalitis**		**Pfizer-BioNTech**	**Moderna**	**Janssen**	**AstraZeneca**	**Sinovac-CoronaVac**	**Sputnik V**	**COVAXIN**	**Total**
		***n* = 0**	***n* = 3**	***n* = 0**	***n* = 0**	***n* = 0**	***n* = 0**	***n* = 0**	***n* = 3**
AGE	<50	0	1(33)	0	0	0	0	0	1(33)
	≥50	0	2(67)	0	0	0	0	0	2(67)
GENDER	M	0	2(67)	0	0	0	0	0	2(67)
	F	0	1(33)	0	0	0	0	0	1(33)
FATAL	Yes	0	0(0)	0	0	0	0	0	0(0)
	No	0	3(100)	0	0	0	0	0	3(100)
**Guillain–Barré Syndrome ^†^**		**Pfizer-BioNTech**	**Moderna**	**Janssen**	**AstraZeneca**	**Sinovac-CoronaVac**	**Sputnik V**	**COVAXIN**	**Total**
		***n* = 3**	***n* = 0**	***n* = 2**	***n* = 9**	***n* = 0**	***n* = 0**	***n* = 0**	***n* = 14**
AGE	<50	0(0)	0	1(50)	2(22)	0	0	0	3(21)
	≥50	3(100)	0	1(50)	7(78)	0	0	0	11(79)
GENDER	M	1(33)	0	1(50)	8(89)	0	0	0	10(71)
	F	2(67)	0	1(50)	1(11)	0	0	0	4(29)
FATAL	Yes	0(0)	0	0(0)	0(0)	0	0	0	0(0)
	No	3(100)	0	2(100)	9(100)	0	0	0	14(100)
**Bell’s Palsy**		**Pfizer-BioNTech**	**Moderna**	**Janssen**	**AstraZeneca**	**Sinovac-CoronaVac**	**Sputnik V**	**COVAXIN**	**Total**
		***n* = 3**	***n* = 0**	***n* = 0**	***n* = 0**	***n* = 1**	***n* = 0**	***n* = 1**	***n* = 5**
AGE	<50	1(33)	0	0	0	1(100)	0	0(0)	2(33)
	≥50	2(67)	0	0	0	0(0)	0	1(100)	3(67)
GENDER	M	2(67)	0	0	0	0(0)	0	1(100)	3(67)
	F	1(33)	0	0	0	1(100)	0	0(0)	2(33)
FATAL	Yes	0(0)	0	0	0	0(0)	0	0(0)	0(0)
	No	3(100)	0	0	0	1(100)	0	1(100)	5(100)
**Others ^††^**		**Pfizer-BioNTech**	**Moderna**	**Janssen**	**AstraZeneca**	**Sinovac-CoronaVac**	**Sputnik V**	**COVAXIN**	**Total**
		***n* = 2**	***n* = 2**	***n* = 0**	***n* = 2**	***n* = 0**	***n* = 0**	***n* = 0**	***n* = 6**
AGE	<50	0(0)	1(50)	0	1(50)	0	0	0	2(33)
	≥50	2(100)	1(50)	0	1(50)	0	0	0	4(67)
GENDER	M	0(0)	2(100)	0	1(50)	0	0	0	3(50)
	F	2(100)	0(0)	0	1(50)	0	0	0	3(50)
FATAL	Yes	0	0(100)	0	0(0)	0	0	0	0(0)
	No	2(100)	2(100)	0	2(100)	0	0	0	6(100)

* Associated with cerebral venous sinus thrombosis (CVST). ** Including acute transverse myelitis (TM), neuromyelitis optica spectrum disorder (NMOSD), multiple sclerosis (MS) exacerbation, acute disseminated encephalomyelitis (ADEM), and longitudinal extensive transverse myelitis (LETM). **^†^** Including chronic inflammatory demyelinating polyradiculoneuropathy (CIDP) and its variants. **^††^** Including small fiber neuropathy, phantosmia, Tolosa–Hunt syndrome, seizure disorders, and acute aseptic meningoencephalitis with Sweet syndrome.

## Data Availability

Data was extracted from the articles published in PUBMED, Google Scholar. This will be provided on request.

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
