# Peer review of "COVID-19 Vaccination and Neurological Manifestations: A Review of Case Reports and Case Series"

_brainsci, 2022, doi:10.3390/brainsci12030407_

Round 1
Reviewer 1 Report
In the review article “COVID-19 Vaccination and Neurological Manifestations: A Re-view of Case Reports and Case Series” Sriwastava et. al., have summarized the neurological side-effects associated with the six most commonly injected COVID-vaccines; Pfizer-BioNTech, Moderna, COVAXIN, AstraZeneca, Sinovac-CoronaVac, Janssen, and Sputnik V. The information assembled in this article advocates for the prevalence of several CNS and PNS manifestations following COVID-19 vaccination, however, most of the cases were non-fatal. Furthermore, authors endorsed that advantage of protection acquired by COVID-19 vaccine is far greater than the hypothesized risk on neurological manifestations. This review is systematically organized with relevant information under appropriate subheadings, figure and tables; hence, it may be accepted for publication after addressing following points.
- As this review includes plenty of acronyms so a paragraph/table describing the expanded form of various acronyms should be added.
- Figure 1 is cropped and incompletely visible in the manuscript draft I downloaded. Please make sure that it is fully displayed in the final version of the article.
- First column of supplementary tables is incompletely visible. Properly formatted tables must be included in the final version of the article to be published.
Plagiarism percentage - not checked by the reviewer.
Author Response
Ref. No.: Manuscript ID: brainsci-1634889
Journal Brain Sciences
Title: COVID-19 Vaccination and Neurological Manifestations: A Review of Case Reports and Case Series
We would like to thank the reviewer for the time they invested in improving the quality of our manuscript. Please see attached point-by-point changes/corrections or explanations to the reviewers’ comments.
In the review article “COVID-19 Vaccination and Neurological Manifestations: A Re-view of Case Reports and Case Series” Sriwastava et. al., have summarized the neurological side-effects associated with the six most commonly injected COVID-vaccines; Pfizer-BioNTech, Moderna, COVAXIN, AstraZeneca, Sinovac-CoronaVac, Janssen, and Sputnik V. The information assembled in this article advocates for the prevalence of several CNS and PNS manifestations following COVID-19 vaccination, however, most of the cases were non-fatal. Furthermore, authors endorsed that advantage of protection acquired by COVID-19 vaccine is far greater than the hypothesized risk on neurological manifestations. This review is systematically organized with relevant information under appropriate subheadings, figure and tables; hence, it may be accepted for publication after addressing following points.
R#1.1. As this review includes plenty of acronyms so a paragraph/table describing the expanded form of various acronyms should be added.
A#1.1. Thank you for the comments. We have now have added abbreviation and foot note. Please see line # 543 to 569 & page # 15-16
R#1.2. Figure 1 is cropped and incompletely visible in the manuscript draft I downloaded. Please make sure that it is fully displayed in the final version of the article.
A#1.2. Thank you for the comments. Please see the updated Figure 1. Please see page # 4
R#1.3. First column of supplementary tables is incompletely visible. Properly formatted tables must be included in the final version of the article to be published.
A#1.3. Thank you for the comments. Please see the updates supplement tables. Please see page # 17 to 33
Reviewer 2 Report
This is an interesting paper about the COVID19 vaccination and neurological manifestations. The paper is well-written and is of interest for the readers. I recommend several minor changes.
The paper is starting with data of the vaccination for the COVID-19. I would recommend to introduce the appearance of the COVID-19 infection, and public health consequences and the development of the COVID-19 vaccines.
In the third paragraph of the introduction section, the authors reported that "neurological complications are reported in several vaccines". Several examples would be helpful.
In the same paragraph the authors reported that the aim of the study is to enumerate post-marketing neurological side-effects of the COVID-19 vaccines. Aims should be described at the end of the introduction section. It would be better to enumerate the vaccines, how and when were developed, and afterwards to describe that the main aim is to review neurological complications applied to each vaccine.
The results section is very extensive. I would recommend to divide the section into many subsections as they categorized cases into the following groups: CNS inflammatory disorders, GBS and its variants, etc.
The discussion section is well-estructured. It has been divided into CNS complications, PNS complications, and Neuroimaging findinds.
I would prefer to present the Figure 1 at the first part of the results section or at the methods section. I will be a good introduction to the topic.
Author Response
Ref. No.: Manuscript ID: brainsci-1634889
Journal Brain Sciences
Title: COVID-19 Vaccination and Neurological Manifestations: A Review of Case Reports and Case Series
We would like to thank the reviewer for the time they invested in improving the quality of our manuscript. Please see attached point-by-point changes/corrections or explanations to the reviewers’ comments.
This is an interesting paper about the COVID19 vaccination and neurological manifestations. The paper is well-written and is of interest for the readers. I recommend several minor changes.
R#2.1. The paper is starting with data of the vaccination for the COVID-19. I would recommend to introduce the appearance of the COVID-19 infection, and public health consequences and the development of the COVID-19 vaccines.
A#2.1. Thank you for your helpful comments. We have now added this relevant information, as suggested, in the Introduction section of our manuscript. Please see line # 55 to 67 & page # 2.
R#2.2. In the third paragraph of the introduction section, the authors reported that "neurological complications are reported in several vaccines". Several examples would be helpful.
A#2.2. Thank you for your helpful comments. We have now added this relevant information. Please see line # 83 to 87 & page # 2.
R#2.3. In the same paragraph the authors reported that the aim of the study is to enumerate post-marketing neurological side-effects of the COVID-19 vaccines. Aims should be described at the end of the introduction section. It would be better to enumerate the vaccines, how and when were developed, and afterwards to describe that the main aim is to review neurological complications applied to each vaccine.
A#2.3. Thank you for this suggestion. We have now have made changes as suggested. Please see line # 112 to 116 & page # 3
R#2.4. The results section is very extensive. I would recommend to divide the section into many subsections as they categorized cases into the following groups: CNS inflammatory disorders, GBS and its variants, etc.
A#2.4. Thank you for this suggestion. We have now divided our findings under CNS and PNS complications (similar to the ‘Discussion’ section) and split the paragraphs under these headings to improve readability. Please page # 6-7
R#2.5. The discussion section is well-structured. It has been divided into CNS complications, PNS complications, and Neuroimaging findings.
A#2.5. Thank you for this comment.
R#2.6. I would prefer to present the Figure 1 at the first part of the results section or at the methods section. I will be a good introduction to the topic.
A#2.6. Thank you for your comment Figure 1 is placed now under method section where the PRISMA is described. Please see page # 4
Reviewer 3 Report
Thank you very much for this interesting article, that reviews a relevant topic I have been often asked about by patients.
The article describes the neurological manifestations that have been described after Covid-Vaccination in a structured way, with the vaccine used and the patient's characteristics. The selection criteria are clear, and the tables provided illustrate the findings well. The context of vaccine averse events and Covid is mentioned in a short and comprehensive way.
I have only a few remarks:
Introduction:
first line states >8bio people vaccinated, which would exceed the world population. Is this not the no of doses administered? Maybe writing the numbers in eg 8.4 billion for better readability
Last paragraph: typo: "We aim is to"
Methods
out of curiosity, I'm wondering if including non English literature would give other information (maybe adverse events are only published in a local language database?). An idea would be to mention it in the discussion (also in relation to the fact that certain vaccines might be overrepresented in certain countries)
Results
p.4 2d paragraph: typo: The most "common?" CNS manifestation was....
Discussion
The first paragraph contains very relevant information, but a clearer structure and better reference to the Table 1a and b could make it more salient (and also: data not available for all the mentioned vaccines?)
References: 50 and 51 seem the same (50 right for authors, 51 for DOI).
Author Response
Ref. No.: Manuscript ID: brainsci-1634889
Journal Brain Sciences
Title: COVID-19 Vaccination and Neurological Manifestations: A Review of Case Reports and Case Series
We would like to thank the reviewer for the time they invested in improving the quality of our manuscript. Please see attached point-by-point changes/corrections or explanations to the reviewers’ comments.
In the review article “COVID-19 Vaccination and Neurological Manifestations: A Review of Case Reports and Case Series” Sriwastava et. al., have summarized the neurological side-effects associated with the six most commonly injected COVID-vaccines; Pfizer-BioNTech, Moderna, COVAXIN, AstraZeneca, Sinovac-CoronaVac, Janssen, and Sputnik V. The information assembled in this article advocates for the prevalence of several CNS and PNS manifestations following COVID-19 vaccination, however, most of the cases were non-fatal. Furthermore, authors endorsed that the advantage of protection acquired by COVID-19 vaccine is far greater than the hypothesized risk on neurological manifestations. This review is systematically organized with relevant information under appropriate subheadings, figures and tables; hence, it may be accepted for publication after addressing the following points.
R#3.1. Introduction: first line states >8bio people vaccinated, which would exceed the world population. Is this not the no of doses administered? Maybe writing the numbers in eg 8.4 billion for better readability. Last paragraph: typo: "We aim is to"
A#3.1. Thank you for these valid corrections. We have made the necessary changes in this section of the paper. Please see line # 54 to 67 & page # 2, & line # 112 & page #2.
R#3.2. Methods: out of curiosity, I'm wondering if including non-English literature would give other information (maybe adverse events are only published in a local language database?). An idea would be to mention it in the discussion (also in relation to the fact that certain vaccines might be overrepresented in certain countries)
A#3.2. Thank you for this insight. This is an interesting point; however, we did not find significant literature in other languages and were not able to incorporate them here.
R#3.3. Results: p.4 2d paragraph: typo: The most "common?" CNS manifestation was....
A#3.3. Thank you for pointing this out. We have now completed this sentence. Please see line # 220 & page #6
R#3.4. Discussion: The first paragraph contains very relevant information, but a clearer structure and better reference to Table 1a and b could make it more salient (and also: data not available for all the mentioned vaccines?)
A#3.4. Thank you for this suggestion. We have made the necessary alterations to improve the structure and included the basis for the difference in data availability across countries and databases. Please see line # 323 to 339 & page # 11
R#3.5. References: 50 and 51 seem the same (50 right for authors, 51 for DOI)
A#3.5. Thank you for bringing this to our attention. We have eliminated the duplicate and corrected the citation. Please see reference # 50 on page # 36
Round 2
Reviewer 1 Report
Authors have addressed all the queries raised by reviewer, so this article should be consider for publication.